# Aqueous Extracts of Four Medicinal Plants and Their Allelopathic Effects on Germination and Seedlings: Their Morphometric Characteristics of Three Horticultural Plant Species

Renata Erhatić [1], Dijana Horvat [1], Zoran Zorić [2], Maja Repajić [2], Tanja Jović [3], Martina Herceg [1], Matea Habuš [1] and Siniša Srečec [1,*]

1   Križevci College of Agriculture, 48260 Križevci, Croatia
2   Faculty of Food Technology and Biotechnology, University of Zagreb, 10000 Zagreb, Croatia
3   Pharmacy Zadar, 23000 Zadar, Croatia
*   Correspondence: ssrecec@vguk.hr

**Abstract:** Allelopathy is a biological phenomenon of synthesizing and excreting compounds that can affect the growth of various organisms, i.e., plant species. The aim of this work was to investigate the possible allelopathic influence of aqueous extracts, obtained from plant species chia (*Salvia hispanica* L.), black cumin (*Nigella sativa* L.), wormwood (*Artemisia absinthium* L.), and nettle (*Urtica dioica* L.), on the seed germination growth and morphometric characteristics of pepper (*Capsicum annuum* L.), spinach (*Spinacia oleracea* L.) and lettuce (*Lactuca sativa* L.) seedlings in laboratory conditions. Pepper, spinach, and lettuce seedlings were treated with aqueous extracts of chia, nettle, wormwood, and black cumin of different concentrations (2.5, 5 and 10%, respectively). The aqueous extracts were prepared according to the method developed by Norsworthy and the qualitative and quantitative analysis of phenolic compounds in aqueous extracts were performed using ultra-performance liquid chromatography electrospray ionization tandem mass spectrometry (UPLC-ESI-MS/MS). Phytochemical composition of chia aqueous extracts showed the highest content (above the 1 mg L$^{-1}$) of epicatechin, quinic acid, caffeic acid, esculetin and cinnamic acid in the comparison with others from the total of 19 detected chemical compounds. In aqueous extracts of black cumin, there were six compounds detected in content above 1 mg L$^{-1}$, i.e., epicatechin, quinic acid, caffeic acid, esculetin, cinnamic acid, and kaempferol. The same compounds were the most abundant in wormwood extracts, along with gallic acid. Epicatechin and esculetin were only two compounds detected in aqueous extract of nettle in concentration higher than 1 mg L$^{-1}$. According to the results of this study, only the treatment of spinach seeds with chia aqueous extract in concentration of 2.5% before germination stimulates the germination of spinach seeds, and wormwood herbs and chia extracts in concentrations of 2.5 and 5% stimulate the germination of lettuce seeds and the growth of hypocotyl and radicle length of developed seedlings. The treatment of pepper and lettuce seeds with aqueous extract of nettle in concentration of 10% completely inhibited seed germination.

**Keywords:** medicinal plants; horticultural plants; aqueous extracts; allelopathy; seeds germination; hypocotyl; radicle; pepper; spinach; lettuce

## 1. Introduction

The term allelopathy is derived from the Greek words *allelon* (one to another) and *pathos* (to suffer), indicating a chemical inhibition of one species towards another. Allelopathy is defined as the chemical interaction of plants through the mediation of chemical compounds secreted by individual plants. Such chemical compounds are called allelochemicals. Allelochemicals are secondary metabolites which might exercise the positive or negative influence of a plant on the growth and development of other plants [1–4]. Essentially, allelopathy is a biological phenomenon by which a plant, fungus or microorganism

produces allelochemicals that, when released into the environment, have an inhibiting or stimulating effect on the growth and development of other organisms [5–7]. Allelochemicals are in fact secondary metabolites, and they act as biostimulators, i.e., promoters or inhibitors of plant growth and development [8,9]. Plant species produce allelochemicals in all tissues and their release plays a significant role in natural ecosystems, in the development of plant communities [10,11] and in agroecosystems [12]. The significance of allelopathy in agricultural production is manifested in the possibility of its application in order to improve soil properties by decomposing plant residues, to increase the diversity of crops by applying it in crop rotation, and to improve the resistance of crops to abiotic factors. However, it is especially important in the control of weeds, diseases and pests, whether directly or by developing biopesticides. The inclusion of allelopathy in integrated or ecological plant protection programs contributes to reducing the use of chemical agents, and thus the occurrence of numerous problems, such as the resistance of an increasing number of pests and pesticide residues, as well as negative effects on the environment and human health. At the same time, it makes it possible to meet the needs of the market for food produced in an environmentally friendly way [13–16]. Traditionally, allelopathy was considered as a bilateral interaction between two plants or plant species. However, allelopathy is now considered as a process within the plant kingdom that can affect and be modulated by other organisms in the plant's environment [17]. Unfortunately, there are no recent studies concerning the allelopathic effects of chia, black cumin, wormwood herbs, and nettle aqueous extracts on seed germination and seedling growth of lettuce, spinach, and pepper.

According to Cheng and Cheng, 2015 [18], the plant allelopathy management practices applied in agriculture are determined by four the most important points: (1) a description of management practices related to allelopathy and allelochemicals in agriculture; (2) a discussion of the progress regarding the mode of action of allelochemicals and the physiological mechanisms of allelopathy, consisting of the influence on cell micro- and ultra-structure, cell division and elongation, membrane permeability, oxidative and antioxidant systems, growth regulation systems, respiration, enzyme synthesis and metabolism, photosynthesis, mineral ion uptake, and protein and nucleic acid synthesis; (3) an evaluation of the effect of ecological mechanisms exerted by allelopathy on microorganisms and the ecological environment; and (4) a discussion of existing problems and proposals for future research directions in this field to provide a useful reference for future studies on plant allelopathy.

The aim of this work was to investigate the possible allelopathic influence of aqueous extracts, obtained from plant species chia (*Salvia hispanica* L.), black cumin (*Nigella sativa* L.), wormwood (*Artemisia absinthium* L.), and nettle (*Urtica dioica* L.), on the seed germination and morphometric characteristics of pepper (*Capsicum annuum* L.), spinach (*Spinacia oleracea* L.) and lettuce (*Lactuca sativa* L.) seedlings in laboratory conditions. Namely, these three species were chosen because of their naturally huge variability of germination rate and because of the huge variability of seedlings hypocotyl and radicle length.

The research was carried out with purpose of assessing the possible application of the mentioned extracts in practice to stimulate the process of the germination and sprouting of seeds, with the assumption that the differences in germination and length of the hypocotyl and radicles of the seedlings would be obtained.

## 2. Materials and Methods

### 2.1. Plant Material and Preparation of Aqueous Extracts

Seeds of chia (*Salvia hispanica* L.) and black cumin (*Nigella sativa* L.), as well as wormwood (*Artemisia absinthium* L.) and nettle (*Urtica dioica* L.) herbs, were used for the preparation of aqueous extracts according to Norsworthy's method, 2003 [19]. The air-dried plant material was cryo-milled with an electric grinder (Retsch GM 200, Retsch GmbH, Haan, Germany) for 13 s at a speed of 10 min$^{-1}$. The equipment was previously cleaned and disinfected in 70% ethanol solution. For the preparation of aqueous extracts of different concentrations (2.5, 5 and 10%) of each plant species, a sufficient amount of crushed mate-

rial was weighed and 250 mL of distilled water was added. The suspensions were mixed and left at room temperature for 24 h. Afterwards, the suspensions were filtered and the obtained supernatants were used for further treatment.

### 2.2. Phytochemical Analyses

Qualitative and quantitative analysis of phenolic compounds in aqueous extracts was performed using UPLC-ESI-MS/MS (Agilent 1290, Agilent, Santa Clara, CA, USA). For the reverse phase separation of compounds, a Fortis C18 column, 1.7 μm, 100 × 2.1 mm (Fortis, Neston, UK) was used according to the method previously described by Elez Garofulić et al. [20]. Analysis of separated compounds was performed by UPLC linked to Agilent 6430 QqQ spectrometer. Ionization was carried out using ESI, in both positive and negative ionization modes, using nitrogen as desolvation and collision gas (Meser, Zapresic, Croatia) at capillary voltage +4/−3.5 kV, a drying gas flow 11 L h$^{-1}$ at 300 °C and nebulizer pressure of 40 psi.

Quantitative determination was performed by the external standard method using the calibration curves of the standards. Quercetin 3-*O*-rhamnoside was determined according to quercetin 3-*O*-rutinoside, epicatechin according to catechin, kaempferol according to kaempferol 3-*O*-glucoside, umbelliferone according to scopoletin, gentisic acid according to protocatechuic acid and *p*-hydroxybenzoic acid according to gallic acid.

All analyses have been performed in triplicate and concentrations of analyzed compounds are expressed as mg L$^{-1}$.

### 2.3. Seed Germination Tests

Certified pepper (*Capsicum annuum* L.), spinach (*Spinacia oleracea* L.) and lettuce (*Lactuca sativa* L.) seeds were disinfected in a 1% solution of sodium hypochlorite (NaOCl) for 20 min and then washed three times with distilled water. Sterile filter paper was placed on the bottom of the sterile Petri dishes, on which 50 seeds were placed. Before placing the seeds on the filter paper, 6 mL of aqueous extract of certain concentrations (2.5, 5 and 10%, respectively) was applied with a pipette. After adding the seeds, the Petri dishes were closed. For each treatment, three replicates and a control without treatment with aqueous extract, i.e., treated only with distilled water, were made. The Petri dishes were placed then in the germination chamber. The germination process for pepper seeds lasted for 14 days, requiring 21 days for spinach and 7 days for lettuce [21].

Germination was calculated according to Equation (1).

$$Germination\ rate\ (\%) = \frac{Number\ of\ germinated\ seeds}{The\ total\ number\ of\ seeds\ in\ the\ Petri\ dish} \times 100 \qquad (1)$$

### 2.4. Morphometric Analyses

To determine the effect of aqueous extracts on morphological characteristics of the seedlings, the morphometric analyses of hypocotyl length and radicle length were conducted according to ISTA methodology for seedling evaluation [21].

### 2.5. Design of the Experiment and Statistical Analysis

The experiment was conducted in laboratory conditions, according to the International Rules for Seed Testing [22]. The seeds of the three horticultural species (pepper, spinach, and lettuce) were treated with four aqueous extracts (of chia, black cumin, wormwood herbs, and nettle) in three concentrations (2.5, 5 and 10%, respectively) in four replications, including the untreated variant per each horticultural species, named as controls and being used for comparison with the treated variants.

The results of chemical analyses of aqueous extracts were represented by descriptive statistics only, while the results of germination tests and morphometric analysis were statistically analyzed by one-way ANOVA and, in the case of a significant F-test ($p < 0.05$), the comparison between the mean values of treatments and control (i.e., not

treated variants) was carried out using the *t* test for dependent samples. In case of a not significant F-test, a *t* test was not conducted. Significant differences were pronounced only in the case of $p < 0.05$ [23]. For statistical analyses, the Statistica TIBCO software was used.

## 3. Results

### 3.1. Results of Phytochemical Analyses of Aqueous Extracts

3.1.1. Chia (*S. hispanica* L.) Aqueous Extracts

The analytical data of phytochemical composition of chia (*S. hispanica* L.) aqueous extracts (Table 1) showed the highest content (above the 1 mg $L^{-1}$) of epicatechin, quinic acid, caffeic acid, esculetin and cinnamic acid in the comparison with others of the total of 19 detected chemical compounds.

**Table 1.** Phytochemical composition of chia (*S. hispanica* L.) aqueous extracts of different concentrations.

| Compound | Extract Concentration | | |
|---|---|---|---|
| | **2.5%** | **5%** | **10%** |
| | **mg $L^{-1}$** | | |
| Epigallocatechin gallate | $0.06 \pm 0.01$ | $0.10 \pm 0.01$ | $0.11 \pm 0.02$ |
| Quercetin 3-*O*-rhamnoside | $0.55 \pm 0.03$ | $0.02 \pm 0.00$ | $0.05 \pm 0.02$ |
| Epicatechin gallate | $0.89 \pm 0.02$ | $0.73 \pm 0.02$ | $0.02 \pm 0.01$ |
| Catechin | $0.18 \pm 0.01$ | $0.05 \pm 0.01$ | $0.06 \pm 0.02$ |
| Epicatechin | $6.70 \pm 0.02$ | $12.35 \pm 0.06$ | $0.05 \pm 0.01$ |
| Kaempferol | $0.91 \pm 0.02$ | $0.81 \pm 0.03$ | $0.79 \pm 0.04$ |
| Naringenin | $0.32 \pm 0.03$ | $0.19 \pm 0.02$ | $0.07 \pm 0.02$ |
| Syringic acid | $0.96 \pm 0.01$ | $0.31 \pm 0.03$ | $0.41 \pm 0.02$ |
| Ferrulic acid | $0.09 \pm 0.02$ | $0.36 \pm 0.05$ | $0.65 \pm 0.04$ |
| Scopoletin | $0.19 \pm 0.01$ | $0.20 \pm 0.01$ | $0.13 \pm 0.03$ |
| Quinic acid | $0.83 \pm 0.02$ | $17.10 \pm 0.01$ | $130.52 \pm 4.57$ |
| Caffeic acid | $2.32 \pm 0.03$ | $1.22 \pm 0.04$ | $0.75 \pm 0.04$ |
| Esculetin | $9.44 \pm 0.05$ | $4.49 \pm 0.01$ | $2.68 \pm 0.20$ |
| Gallic acid | $0.55 \pm 0.01$ | $0.50 \pm 0.02$ | $0.53 \pm 0.06$ |
| *p*-Coumaric acid | $0.07 \pm 0.01$ | $0.07 \pm 0.01$ | $0.20 \pm 0.05$ |
| Umbelliferone | $0.25 \pm 0.02$ | $0.14 \pm 0.01$ | $0.71 \pm 0.10$ |
| Gentisic acid | $0.05 \pm 0.01$ | $0.04 \pm 0.01$ | $0.11 \pm 0.03$ |
| Protocatechuic acid | $0.07 \pm 0.01$ | $0.02 \pm 0.01$ | $0.09 \pm 0.01$ |
| Cinnamic acid | $7.12 \pm 0.04$ | $7.78 \pm 0.02$ | $4.84 \pm 0.19$ |
| *p*-Hydroxybenzoic acid | $0.96 \pm 0.03$ | $0.82 \pm 0.04$ | $0.87 \pm 0.09$ |

Results are expressed as mean $\pm$ standard deviation (SD).

3.1.2. Black Cumin (*N. sativa* L.) Aqueous Extracts

In aqueous extracts of black cumin (*N. sativa* L.), six compounds were detected with content higher than 1 mg $L^{-1}$ (Table 2), namely epicatechin, quinic acid, caffeic acid, esculetin, cinnamic acid and kaempferol.

3.1.3. Wormwood Herbs (*A. absinthium* L.) Aqueous Extracts

The results of phytochemical analysis of wormwood herb (*A. absinthium* L.) aqueous extracts (Table 3) showed almost the same order of compounds in the extracts with content higher than 1 mg $L^{-1}$ as in aqueous extracts of black cumin, with the exception of gallic acid content. This means that the highest content of epicatechin was also detected in aqueous extracts of wormwood herbs, followed by esculetin, cinnamic acid, quinic acid, kaempferol and finally gallic acid.

**Table 2.** Phytochemical composition of black cumin (*N. sativa* L.) aqueous extracts of different concentrations.

| Compound | Extract Concentration | | |
|---|---|---|---|
| | **2.5%** | **5%** | **10%** |
| | mg L$^{-1}$ | | |
| Epigallocatechin gallate | 0.06 ± 0.02 | 0.07 ± 0.03 | 0.08 ± 0.03 |
| Quercetin 3-*O*-rhamnoside | 0.06 ± 0.03 | 0.05 ± 0.05 | 0.03 ± 0.02 |
| Epicatechin gallate | 0.30 ± 0.06 | 0.86 ± 0.10 | 0.71 ± 0.21 |
| Catechin | 0.04 ± 0.02 | 0.08 ± 0.07 | 0.10 ± 0.06 |
| Epicatechin | 7.08 ± 0.36 | 11.78 ± 1.32 | 13.52 ± 1.21 |
| Kaempferol | 1.40 ± 0.31 | 1.76 ± 0.22 | 0.91 ± 0.10 |
| Naringenin | 0.23 ± 0.06 | 0.20 ± 0.09 | 0.30 ± 0.11 |
| Syringic acid | 0.39 ± 0.08 | 0.82 ± 0.14 | 0.37 ± 0.09 |
| Ferrulic acid | 0.17 ± 0.07 | 0.13 ± 0.03 | 0.43 ± 0.09 |
| Scopoletin | 0.69 ± 0.13 | 0.20 ± 0.01 | 0.39 ± 0.17 |
| Quinic acid | 0.79 ± 0.10 | 1.45 ± 0.32 | 8.82 ± 0.28 |
| Caffeic acid | 2.36 ± 0.08 | 5.22 ± 0.09 | 5.37 ± 0.13 |
| Esculetin | 4.47 ± 0.34 | 6.57 ± 0.21 | 4.87 ± 0.36 |
| Gallic acid | 0.76 ± 0.12 | 0.67 ± 0.24 | 0.60 ± 0.15 |
| *p*-Coumaric acid | 0.10 ± 0.04 | 0.09 ± 0.04 | 0.06 ± 0.02 |
| Umbelliferone | 0.51 ± 0.22 | 0.52 ± 0.07 | 0.21 ± 0.06 |
| Gentisic acid | 0.05 ± 0.04 | 0.03 ± 0.02 | 0.05 ± 0.04 |
| Protocatechuic acid | 0.06 ± 0.04 | 0.03 ± 0.03 | 0.09 ± 0.07 |
| Cinnamic acid | 1.24 ± 0.08 | 0.64 ± 0.10 | 1.35 ± 0.39 |
| *p*-Hydroxybenzoic acid | 0.90 ± 0.09 | 1.30 ± 0.15 | 0.93 ± 0.08 |

Results are expressed as mean ± SD.

**Table 3.** Phytochemical composition of wormwood herbs (*A. absinthium* L.) aqueous extracts of different concentrations.

| Compound | Extract Concentration | | |
|---|---|---|---|
| | **2.5%** | **5%** | **10%** |
| | mg L$^{-1}$ | | |
| Epigallocatechin gallate | 0.09 ± 0.02 | 0.06 ± 0.01 | 0.07 ± 0.02 |
| Quercetin 3-*O*-rhamnoside | 0.02 ± 0.01 | 0.04 ± 0.02 | 0.05 ± 0.02 |
| Epicatechin gallate | 0.76 ± 0.07 | 0.64 ± 0.23 | 0.23 ± 0.06 |
| Catechin | 0.07 ± 0.03 | 0.11 ± 0.08 | 0.09 ± 0.08 |
| Epicatechin | 19.48 ± 0.96 | 20.18 ± 1.43 | 18.36 ± 1.65 |
| Kaempferol | 1.72 ± 0.09 | 1.18 ± 0.37 | 1.27 ± 0.12 |
| Naringenin | 0.37 ± 0.06 | 0.18 ± 0.12 | 0.17 ± 0.11 |
| Syringic acid | 0.93 ± 0.12 | 0.62 ± 0.16 | 0.65 ± 0.20 |
| Ferrulic acid | 0.30 ± 0.02 | 0.14 ± 0.05 | 0.24 ± 0.08 |
| Scopoletin | 0.48 ± 0.02 | 0.47 ± 0.07 | 0.47 ± 0.21 |
| Quinic acid | 1.00 ± 0.09 | 6.43 ± 0.60 | 7.04 ± 0.72 |
| Caffeic acid | 0.09 ± 0.04 | 1.30 ± 0.05 | 0.27 ± 0.40 |
| Esculetin | 2.36 ± 0.26 | 2.23 ± 0.55 | 13.17 ± 1.12 |
| Gallic acid | 1.10 ± 0.11 | 1.07 ± 0.12 | 0.75 ± 0.11 |
| *p*-Coumaric acid | 0.09 ± 0.02 | 0.18 ± 0.02 | 0.09 ± 0.01 |
| Umbelliferone | 0.28 ± 0.17 | 0.33 ± 0.09 | 0.48 ± 0.07 |
| Gentisic acid | 0.02 ± 0.01 | 0.04 ± 0.02 | 0.04 ± 0.01 |
| Protocatechuic acid | 0.04 ± 0.01 | 0.04 ± 0.03 | 0.03 ± 0.03 |
| Cinnamic acid | 2.77 ± 0.11 | 4.83 ± 0.23 | 8.20 ± 0.23 |
| *p*-Hydroxybenzoic acid | 0.59 ± 0.51 | 1.43 ± 0.14 | 1.16 ± 0.18 |

Results are expressed as mean ± SD.

### 3.1.4. Nettle (*U. dioica* L.) Aqueous Extracts

The phytochemical composition of nettle (*U. dioica* L.) aqueous extracts (Table 4) was different than the phytochemical composition of aqueous extracts of previous plant species. Namely, only two compounds with a content higher than 1 mg $L^{-1}$ were detected in aqueous extracts of different concentrations, i.e., epicatechin and esculetin.

**Table 4.** Phytochemical composition of nettle (*U. dioica* L.) aqueous extracts of different concentrations.

| Compound | Extract Concentration | | |
|---|---|---|---|
| | **2.5%** | **5%** | **10%** |
| | **mg $L^{-1}$** | | |
| Epigallocatechin gallate | 0.09 ± 0.02 | 0.07 ± 0.02 | 0.09 ± 0.02 |
| Quercetin 3-*O*-rhamnoside | 0.04 ± 0.02 | 0.74 ± 0.11 | 0.93 ± 0.10 |
| Epicatechin gallate | 0.77 ± 0.15 | 0.47 ± 0.21 | 0.89 ± 0.15 |
| Catechin | 0.14 ± 0.03 | 0.03 ± 0.02 | 0.24 ± 0.06 |
| Epicatechin | 18.50 ± 1.78 | 14.45 ± 1.87 | 19.99 ± 2.06 |
| Kaempferol | 0.94 ± 0.21 | 0.89 ± 0.16 | 0.96 ± 0.27 |
| Naringenin | 0.19 ± 0.10 | 1.19 ± 0.03 | 0.11 ± 0.02 |
| Syringic acid | 0.23 ± 0.07 | 0.30 ± 0.09 | 0.15 ± 0.01 |
| Ferrulic acid | 0.25 ± 0.04 | 0.17 ± 0.04 | 0.67 ± 0.12 |
| Scopoletin | 0.54 ± 0.06 | 0.34 ± 0.26 | 0.30 ± 0.16 |
| Quinic acid | 0.93 ± 0.10 | 8.87 ± 0.79 | 10.67 ± 0.39 |
| Caffeic acid | 0.04 ± 0.04 | 0.09 ± 0.09 | 0.76 ± 0.26 |
| Esculetin | 5.11 ± 0.71 | 4.44 ± 0.43 | 7.15 ± 0.62 |
| Gallic acid | 0.77 ± 0.19 | 0.49 ± 0.06 | 0.57 ± 0.05 |
| *p*-Coumaric acid | 0.07 ± 0.01 | 0.09 ± 0.02 | 0.06 ± 0.00 |
| Umbelliferone | 0.30 ± 0.12 | 0.16 ± 0.06 | 0.49 ± 0.06 |
| Gentisic acid | 0.03 ± 0.02 | 0.03 ± 0.02 | 0.05 ± 0.04 |
| Protocatechuic acid | 0.03 ± 0.03 | 0.02 ± 0.01 | 0.02 ± 0.01 |
| Cinnamic acid | 0.88 ± 0.28 | 0.94 ± 0.17 | 6.13 ± 0.37 |
| *p*-Hydroxybenzoic acid | 1.05 ± 0.19 | 1.02 ± 0.22 | 0.99 ± 0.15 |

Results are expressed as mean ± SD.

All the analytical data of phytochemical analyses are available in Table S1 in Supplementary Materials.

### 3.2. Results of Germination Tests and Morphometric Analyses of Seedlings

#### 3.2.1. Germination Tests

Pepper seed (*C. annuum* L.) germination was lower in all treatments with plant aqueous extracts in comparison with the control, i.e., untreated variant (Table 5). Moreover, the nettle (*U. dioica* L.) aqueous extract at 10% concentration completely inhibited the germination of pepper seeds (Table 5, Figures A1–A4) and germination of pepper seeds was lower in comparison with the control in all variants treated with aqueous extracts of wormwood herbs, black cumin and chia plants at concentrations of 10%.

**Table 5.** Comparison of differences of means for pepper seeds germination (*C. annuum* L.) (The values in the table represent the mean value of germination rate in % ± SD).

| Control (Not Treated) | 71.5 ± 5.74 | | | |
|---|---|---|---|---|
| **Treatments** | **Chia (*S. hispanica* L.)** | **Black Cumin (*N. sativa* L.)** | **Wormwood Herbs (*A. absinthium* L.)** | **Nettle (*U. dioica* L.)** |
| 2.5% | 61.0 ± 6.00 [a] | 64.0 ± 9.52 [a] | 64.5 ± 5.74 [b] | 55.5 ± 11.70 [a] |
| 5% | 66.5 ± 3.41 [b] | 64.0 ± 10.95 [a] | 57.5 ± 16.44 [a] | 43.5 ± 13.30 [b] |
| 10% | 65.5 ± 2.51 [b] | 52.5 ± 5.97 [b] | 31.5 ± 15.17 [c] | 0.0 ± 0.00 |

The values with different letters in superscript are significant $p < 0.05$.

Despite the huge decrease in germination rates in all treated variants in the comparison with control, i.e., not treated variant, it is obvious that only few differences were statistically significant. Namely, in some cases the variability between the replications was higher than the variability between the treatments, which caused a significant F-test factor after one-way ANOVA. In cases of not significant F-test, the *t* test was not calculated. Nevertheless, that the decrease in germination includes malformations of the seedlings is quite visible from Figures A1–A4 in Appendix A.

The results of spinach (*S. oleracea* L.) seed germination correspond with the results of pepper seeds germination (Table 6), with the exception that a concentration of 10% nettle aqueous extract did not stop the germination process like in previous cases (Figures A5–A8 in Appendix A).

**Table 6.** Comparison of differences of means for spinach seeds germination (*S. oleracea* L.) (The values in the table represent the mean value of germination rate in % ± SD).

| Control (Not Treated) | 93.0 ± 2.58 | | | |
|---|---|---|---|---|
| **Treatments** | **Chia** (*S. hispanica* L.) | **Black Cumin** (*N. sativa* L.) | **Wormwood Herbs** (*A. absinthium* L.) | **Nettle** (*U. dioica* L.) |
| 2.5% | 97 ± 1.15 [b] | 91.5 ± 5.74 [n.s.] | 90 ± 7.12 [n.s.] | 92.5 ± 3.41 [n.s.] |
| 5% | 90.0 ± 2.83 [a] | 90.5 ± 10.25 [n.s.] | 93 ± 1.11 [n.s.] | 90.5 ± 9.98 [n.s.] |
| 10% | 92 ± 4.89 [c] | 94.0 ± 2.83 [n.s.] | 86.5 ± 5.97 [n.s.] | 94.5 ± 1.91 [n.s.] |

The values with different letters in superscript are significant $p < 0.05$; (n.s. = not significant).

However, in Table 6 a certain increase in germination rate in variants of treatments with 10% aqueous extract of nettle and black cumin can be observed. Nevertheless, these values are not statistically significant (i.e., $p > 0.05$).

The results of lettuce (*L. sativa* L.) seed germination are almost contradictory with the results of seeds germination of the treated variants of previous horticultural plant species. The variants treated with 2.5% aqueous extract of wormwood herbs and chia showed higher germination rate in the comparison with the control (Table 7). However, the treatment with a 5% aqueous extract of nettle significantly decreased the germination, while a 10% aqueous extract of nettle completely inhibited germination. At the same time, all variants treated with all the concentrations of aqueous extracts of wormwood herbs, black cumin and chia showed higher germination rates in the comparison with the control (Table 7). This is also visible in Figures A9–A12 in Appendix A.

**Table 7.** Comparison of differences of means for lettuce seeds germination (*L. sativa* L.) (The values in the table represent the mean value of germination rate in % ± SD).

| Control (Not Treated) | 86.5 ± 5.74 | | | |
|---|---|---|---|---|
| **Treatments** | **Chia** (*S. hispanica* L.) | **Black Cumin** (*N. sativa* L.) | **Wormwood Herbs** (*A. absinthium* L.) | **Nettle** (*U. dioica* L.) |
| 2.5% | 98.5 ± 1.91 [a] | 94 ± 1.63 [n.s.] | 99 ± 1.15 [a] | 95 ± 4.16 [a] |
| 5% | 95 ± 2.58 [c] | 93 ± 2.00 [n.s.] | 94 ± 4.32 [b] | 16 ± 5.16 [b] |
| 10% | 96 ± 1.63 [b] | 91.5 ± 8.69 [n.s.] | 91 ± 5.03 [c] | 0.0 ± 0.00 |

The values with different letters in superscript are significant $p < 0.05$; (n.s. = not significant).

### 3.2.2. Morphometric Analyses of Seedlings

Hypocotyl Length

The hypocotyl length of pepper (*C. annuum* L.) seedlings was significantly smaller in the comparison with the control (Table 8).

**Table 8.** Comparison of differences of means hypocotyl length of pepper (*C. annuum* L.) seedlings (The values in the table represent the mean value of hypocotyl length in cm $\pm$ SD).

| Control (Not Treated) | 3.73 $\pm$ 0.29 | | | |
|---|---|---|---|---|
| **Treatments** | **Chia** (*S. hispanica* L.) | **Black Cumin** (*N. sativa* L.) | **Wormwood Herbs** (*A. absinthium* L.) | **Nettle** (*U. dioica* L.) |
| 2.5% | 2.225 $\pm$ 0.25 [b] | 2.20 $\pm$ 0.42 [b] | 2.47 $\pm$ 0.26 [a] | 1.85 $\pm$ 0.24 [a] |
| 5% | 2.42 $\pm$ 0.05 [c] | 2.37 $\pm$ 0.09 [c] | 1.90 $\pm$ 0.14 [b] | 1.73 $\pm$ 0.15 [b] |
| 10% | 2.17 $\pm$ 0.22 [a] | 2.075 $\pm$ 0.34 [a] | 1.72 $\pm$ 0.17 [c] | 0.0 $\pm$ 0.00 |

The values with different letters in superscript are significant $p < 0.05$.

On the other hand, the hypocotyl length of spinach (*S. oleracea* L.) seedlings was significantly higher in variants treated with 2.5 and 5% concentrations of nettle aqueous extract, as well as with 2.5 and 5% concentrations of black cumin aqueous extract. The other differences were not significant (Table 9).

**Table 9.** Comparison of differences of means for hypocotyl length of spinach (*S. oleracea* L.) seedlings (The values in the table represent the mean value of hypocotyl length in cm $\pm$ SD).

| Control (Not Treated) | 3.40 $\pm$ 0.41 | | | |
|---|---|---|---|---|
| **Treatments** | **Chia** (*S. hispanica* L.) | **Black Cumin** (*N. sativa* L.) | **Wormwood Herbs** (*A. absinthium* L.) | **Nettle** (*U. dioica* L.) |
| 2.5% | 4.12 $\pm$ 0.37 [n.s.] | 4.55 $\pm$ 0.48 [a] | 3.72 $\pm$ 0.22 [n.s.] | 4.45 $\pm$ 0.13 [a] |
| 5% | 3.27 $\pm$ 0.43 [n.s.] | 4.57 $\pm$ 0.26 [a] | 3.60 $\pm$ 0.16 [n.s.] | 4.55 $\pm$ 0.40 [a] |
| 10% | 3.00 $\pm$ 0.41 [n.s.] | 3.60 $\pm$ 0.63 [b] | 3.32 $\pm$ 0.68 [n.s.] | 3.10 $\pm$ 0.40 [b] |

The values with different letters in superscript are significant $p < 0.05$; (n.s. = not significant).

The treatments with wormwood aqueous extracts of 2.5 and 5% concentration, black cumin aqueous extract of 2.5% concentration, as well as with chia aqueous extract of 2.5, 5 and 10% concentration, respectively, increased the hypocotyl length of lettuce (*L. sativa* L.). The comparisons between the other variants were not significant (Table 10).

**Table 10.** Comparison of differences of means for hypocotyl length of lettuce (*L. sativa* L.) seedlings (The values in the table represent the mean value of hypocotyl length in cm $\pm$ SD).

| Control (Not Treated) | 2.30 $\pm$ 0.27 | | | |
|---|---|---|---|---|
| **Treatments** | **Chia** (*S. hispanica* L.) | **Black Cumin** (*N. sativa* L.) | **Wormwood Herbs** (*A. absinthium* L.) | **Nettle** (*U. dioica* L.) |
| 2.5% | 3.72 $\pm$ 0.66 [b] | 3.95 $\pm$ 0.17 [c] | 4.0 $\pm$ 0.32 [c] | 3.5 $\pm$ 0.6 [a] |
| 5% | 4.55 $\pm$ 0.57 [c] | 3.15 $\pm$ 0.33 [a] | 3.30 $\pm$ 0.14 [b] | 2.05 $\pm$ 1.37 [a] |
| 10% | 3.27 $\pm$ 0.40 [a] | 3.25 $\pm$ 0.52 [b] | 2.22 $\pm$ 0.43 [a] | 0.0 $\pm$ 0.00 |

The values with different letters in superscript are significant $p < 0.05$.

Radicle Length

The length of pepper (*C. annuum* L.) seedlings radicle (Table 11) completely corresponds with the results of the length of the hypocotyl length (Table 8). Nevertheless, it is visible from the results presented in Table 11 that only four comparisons between control and treated variants were significant.

**Table 11.** Comparison of differences of means radicle length of pepper (*C. annuum* L.) seedlings (The values in the table represent the mean value of radicle length in cm $\pm$ SD).

| Control (Not Treated) | | $2.55 \pm 0.24$ | | |
|---|---|---|---|---|
| **Treatments** | **Chia** (*S. hispanica* L.) | **Black Cumin** (*N. sativa* L.) | **Wormwood Herbs** (*A. absinthium* L.) | **Nettle** (*U. dioica* L.) |
| 2.5% | $2.30 \pm 0.43$ [n.s.] | $2.42 \pm 0.26$ [a] | $2.27 \pm 0.67$ [a] | $2.25 \pm 0.30$ [a] |
| 5% | $2.17 \pm 0.39$ [n.s.] | $1.65 \pm 0.13$ [b] | $2.5 \pm 0.14$ [a] | $1.15 \pm 0.38$ [b] |
| 10% | $2.15 \pm 0.24$ [n.s.] | $1.32 \pm 0.57$ [c] | $1.55 \pm 0.49$ [b] | $0.0 \pm 0.00$ |

The values with different letters in superscript are significant $p < 0.05$; (n.s. = not significant).

The comparisons of radicle length between control and treated variants of spinach (*S. oleracea* L.) seedlings (Table 12) in certain measures correspond with the results of the spinach first foliar leaf length (Table 9). However, because of the presence of only one significant comparison, the results revealed in Table 12 must be taken with the certain reserve.

**Table 12.** Comparison of differences of means for radicle length of spinach (*S. oleracea* L.) seedlings (The values in the table represent the mean value of radicle length in cm $\pm$ SD).

| Control (Not Treated) | | $4.90 \pm 1.34$ | | |
|---|---|---|---|---|
| **Treatments** | **Chia** (*S. hispanica* L.) | **Black Cumin** (*N. sativa* L.) | **Wormwood Herbs** (*A. absinthium* L.) | **Nettle** (*U. dioica* L.) |
| 2.5% | $5.92 \pm 0.74$ [n.s.] | $5.10 \pm 1.07$ [a] | $5.00 \pm 1.08$ [n.s.] | $3.92 \pm 0.40$ [n.s.] |
| 5% | $5.32 \pm 0.86$ [n.s.] | $3.67 \pm 0.88$ [a] | $4.07 \pm 0.46$ [n.s.] | $3.32 \pm 1.13$ [n.s.] |
| 10% | $4.30 \pm 0.61$ [n.s.] | $2.10 \pm 0.27$ [b] | $2.40 \pm 0.42$ [n.s.] | $2.85 \pm 0.4$ [n.s.] |

The values with different letters in superscript are significant $p < 0.05$; (n.s. = not significant).

On the other hand, the results of lettuce (*L. sativa* L.) seedlings radicle length (Table 13) correspond with the results of their hypocotyl length (Table 10).

**Table 13.** Comparison of differences of means for radicle length of lettuce (*L. sativa* L.) seedlings (The values in the table represent the mean value of radicle length in cm $\pm$ SD).

| Control (Not Treated) | | $1.77 \pm 0.34$ | | |
|---|---|---|---|---|
| **Treatments** | **Chia** (*S. hispanica* L.) | **Black Cumin** (*N. sativa* L.) | **Wormwood Herbs** (*A. absinthium* L.) | **Nettle** (*U. dioica* L.) |
| 2.5% | $2.92 \pm 0.67$ [a] | $3.37 \pm 0.22$ [a] | $3.05 \pm 0.45$ [n.s.] | $1.67 \pm 0.30$ [a] |
| 5% | $2.98 \pm 1.08$ [b] | $2.47 \pm 0.59$ [b] | $2.35 \pm 0.31$ [n.s.] | $0.57 \pm 0.40$ [b] |
| 10% | $2.25 \pm 0.34$ [b] | $1.77 \pm 0.15$ [b] | $1.30 \pm 0.01$ [n.s.] | $0.0 \pm 0.00$ |

The values with different letters in superscript are significant $p < 0.05$; (n.s. = not significant).

All the analytical data of germination tests and morphometric analyses are available in Table S2 in Supplementary Material.

## 4. Discussion

The results of lettuce seeds germination, after treatment with aqueous extracts of chia, black cumin, wormwood herbs, and nettle (Table 7), as well as hypocotyl and radicle length of lettuce seedlings (Tables 10 and 13), partially correspond with the results of previous studies [24,25]. The authors of previous studies found an inhibitory effects of epicatechin on the growth of lettuce seedlings. However, in the case when epicatechin is applied with hydroquinone, the epicatechin tends to counteract the growth inhibition activity of hydroquinone [26]. Moreover, the application of chia aqueous extracts in all concentrations, i.e., of 2.5, 5 and 10%, respectively, significantly increased the lettuce seeds germination in comparison with the control (Table 7). On the other hand, only one variant of spinach seeds treated with aqueous extract of chia saw a significant increased in the spinach

seeds germination (Table 6), although all the treatments with all aqueous extracts in all concentrations significantly decreased the pepper seeds germination (Table 5). Additionally, when comparing these results with the results of hypocotyl and radicle lengths of pepper seedlings in the same variants of trial, it is visible that the decrease in hypocotyl (Table 8) and radicle length (Table 11) corresponds with decrease in the germination of pepper seeds (Table 5). The correspondence between the results of lettuce seeds germination and lengths of lettuce seedlings hypocotyl and radicle also occurs (Tables 7, 10 and 13), but there is no correspondence between the spinach seeds germination and lengths of hypocotyl and radicle of spinach seedlings (Tables 6, 9 and 12). The results of some studies show a strong negative correlation between the increasing of concentrations of volatile oils apart the pure components limonene and cineole purified from *Eucalyptus globulus* on seed germination and vigour [27]. Moreover, the increasing doses of *Vicia sativa* aqueous extracts caused decrease in germination of mungbean and mashbean [28] and the aqueous extracts of some weeds decreased the germination of wheat and chickpea seeds [29].

Nevertheless, the results of this study, at the first glance, seem rather contradictory. For instance, it is obvious that wormwood aqueous herbs extracts in 2.5 and 5% concentrations significantly increased the germination of lettuce seeds (Table 7).

A question remains: why is that so? This is a very easy but, at the same time, tough question, considering the fact that all the treatments were performed with the same aqueous extracts, and prepared from the same lots of nettles, wormwood herbs, black cumin, and chia in the same concentrations. The same is true for the seeds of pepper, spinach and lettuce, which also belong to the same lots. Moreover, the experimental conditions, such as moisture and temperature, were completely controlled since the trials were conducted in the same laboratory by the same staff. Besides, the phytochemical analyses of aqueous extracts showed the highest content (above the 1 mg $L^{-1}$) of epicatechin, quinic acid, caffeic acid, esculetin and cinnamic acid in chia, and with those compounds kaemferol was additionally detected in aqueous extracts of black cumin and wormwood. With the previously mentioned compounds, gallic acid was detected in content higher than 1 mg $L^{-1}$ (Tables 1–3 and S1) in wormwood extracts. On the other hand, in nettle aqueous extracts, there were only two compounds present in concentrations higher than 1 mg $L^{-1}$, namely epicatechin and esculetin (Tables 4 and S1). Thus, considering the previously mentioned certain contradiction in the achieved results, it is important to point out that similar results were obtained in some legumes at the beginning of the 1980s and that the authors of that study found the combination of some phenols, gallic acid, β-naphthol and gibberellic acid to show no influence on germination but a significant influence on seedling growth [30]. This corresponds with the fact that treatment with 10 mM of gibberellic acid might increase the amino acid and protein synthesis in the cells of the meristems, developing vascular tissues of the embryonic axis and cotyledons in the seeds of *Sinapis arvensis* [31]. Besides, the results of recent studies show that treatment with 2 μM of α-naphthalene acetic acid cause a high induction of *Stevia rebaudiana* calluses and increase their biomass [32]. Although, some polyphenolic rich medicinal plants cause a decrease in seed germination and seedling growth of many cultivated plants [33].

The physiological role of phenolic substances seemed to be pretty controversial, and the researchers were mostly divided about their importance. Moreover, it was the prevailing opinion that exogenously applied phenols depress the growth of plant tissues, but at the same time, that exogenously applied phenols might modify the growth response to gibberellins and cytokinins [34]. It is well known that an increase in seed respiration strictly corresponds to germination process. That includes processes such as glycolysis, the oxidative pentose phosphate pathway, the tricarboxylic cycle and finally oxidative phosphorylation [35,36]. The treatment of seeds with phenolic compounds strongly correlated to the inhibition of enzymes of glycolysis and the oxidative pentose phosphate pathway, which might be a possible explanation for the decrease in seed germination [37]. According to some authors, the synergistic effect between the phenolic compounds and plant hormones is not excluded. This is because the phenolic acids might show hormone-like activity and

consequently be able to act through the plant hormones as biostimulants [38]. It is very important to emphasize that the studies concerned the allelopathic potential of lettuce and pepper. Namely, the lettuce extracts significantly delayed the root growth of alfalfa and barnyard grass. The extract of lettuce showed allelopathic potential on seed germination and seedlings growth [39–41]. The low molecular weight of phenolic compounds isolated from pepper leaves, and from the soil in which the pepper was decomposed, showed an inhibition of seed germination of *Chenopodium album* L., *Plantago lanceolata* L., *Amaranthus retroflexus* L., *Solanum nigrum* L., *Cirsium* sp. and *Rumex crispus* L. [42].

## 5. Conclusions

Considering the achieved results, it remains to be asked: what are the benefits and harms of treatments of pepper (*C. annuum* L.), spinach (*S. olereacea* L.), and lettuce (*L. sativa* L.) seeds with aqueous extracts of chia (*S. hispanica* L.), black cumin (*N. sativa* L.), wormwood herbs (*A. absinthium* L.), and nettle (*U. dioica* L.)?

According to the results of this study, all the aqueous extracts in all applied concentrations of investigated medicinal plants caused significant decrease in pepper (*C. annuum* L.) seed germination. Moreover, treatment with 10% aqueous extract of nettle showed a complete inhibition effect on pepper seed germination. The length of hypocotyl and radicle of pepper seedlings developed from sprouted pepper seeds was also smaller in comparison with the control. Chia (*S. hispanica* L.) 2.5% aqueous extract increased the germination of spinach (*S. olereacea* L.) seeds, but the differences in hypocotyl and radicle length of seedlings of the same variant of treatment were not significant. The variants treated with 2.5 and 5% aqueous extract of wormwood herbs (*A. absinthium* L.) and chia (*S. hispanica* L.) increased the germination of lettuce seeds in the comparison with the control, as well as hypocotyl and radicle length of developed seedlings. The 10% aqueous extract of nettle completely inhibited the germination of lettuce seeds, much like the same phenomenon noted in spinach seed germination tests.

Thus, only the treatment of spinach seeds before germination with chia aqueous extract in a concentration of 2.5% stimulated germination of spinach seeds and, wormwood herbs and chia extracts in concentrations of 2.5 and 5% stimulated the germination of lettuce seeds and the growth of hypocotyl and radicle length of developed seedlings. Taking into account the absence and/or insufficient number of studies concerning the allelopathic effect of investigated medicinal plant species aqueous extracts on germination and morphometric traits of pepper, lettuce, and spinach seedlings, the synergistic or antagonistic effects of some phytochemicals are not excluded. Such a topic may be elucidated by future research.

**Supplementary Materials:** The following supporting information can be downloaded at: https://www.mdpi.com/article/10.3390/app13042258/s1, Table S1: Phytochemical characterization of plant aqueous extracts; Table S2: Germination rates and morphometric characteristics of pepper, spinach and lettuce seedlings.

**Author Contributions:** Conceptualization, D.H., R.E. and S.S.; methodology, R.E., D.H. and S.S.; validation, D.H., R.E. and S.S.; formal analysis, M.H. (Martina Herceg), Z.Z. and T.J.; investigation, R.E., D.H. and S.S.; resources, M.R.; data curation, S.S. and Z.Z.; writing—original draft preparation, S.S.; writing—review and editing, M.H. (Matea Habuš), M.R., Z.Z. and S.S.; visualization, M.H. (Matea Habuš) and M.R. All authors have read and agreed to the published version of the manuscript.

**Funding:** This research was funded by the Križevci College of Agriculture and by the Croatian Science Foundation, research project entitled as "Isolation and encapsulation of bioactive molecules of wild and cultivated nettle and fennel and effects on organism physiology". Project code: HRZZ IP-01-2018-4924.

**Institutional Review Board Statement:** Not applicable. Humans and animals were not involved in this study.

**Informed Consent Statement:** Not applicable. Humans and animals were not involved in this study.

**Data Availability Statement:** Not applicable. The authors used only their own experimental data.

**Conflicts of Interest:** The authors declare no conflict of interest and the funders had no role in the design of the study; in the collection, analyses, or interpretation of data; in the writing of the manuscript; or in the decision to publish the results.

## Appendix A

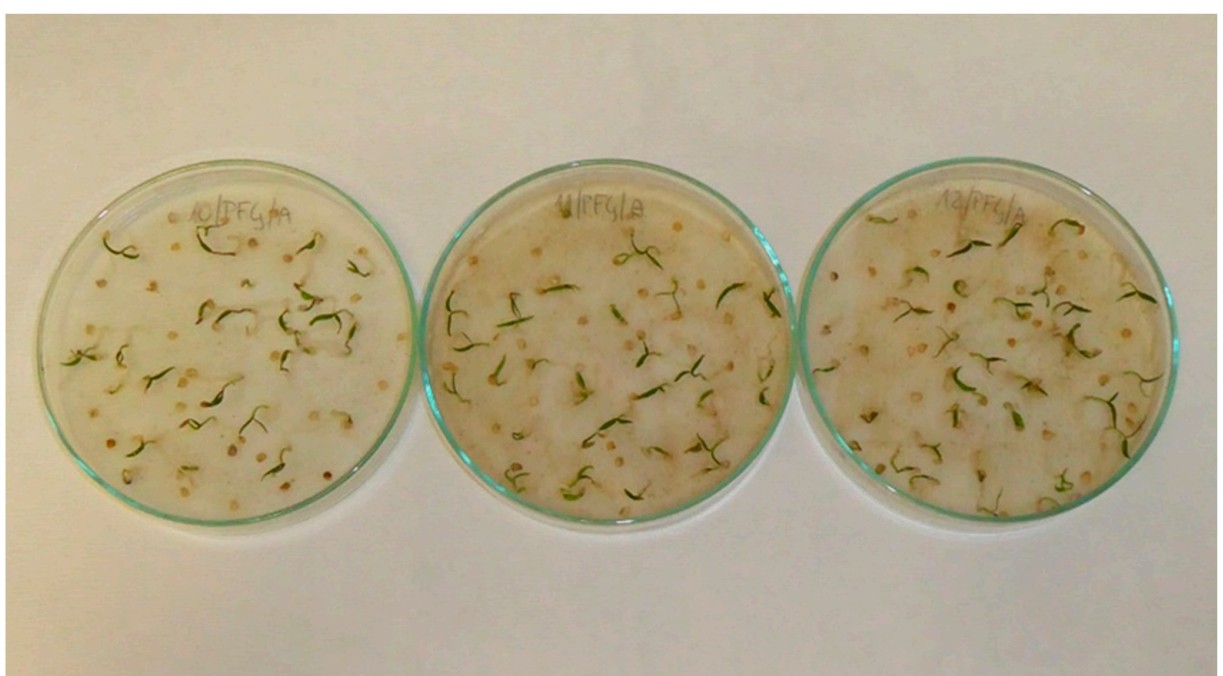

**Figure A1.** Pepper (*Capsicum anuum* L.) seedlings; treatments with aqueous extracts of chia (*Salvia hispanica* L.) in 2.5% concentration (**left**), 5% concentration (**middle**) and 10% concentration (**right**).

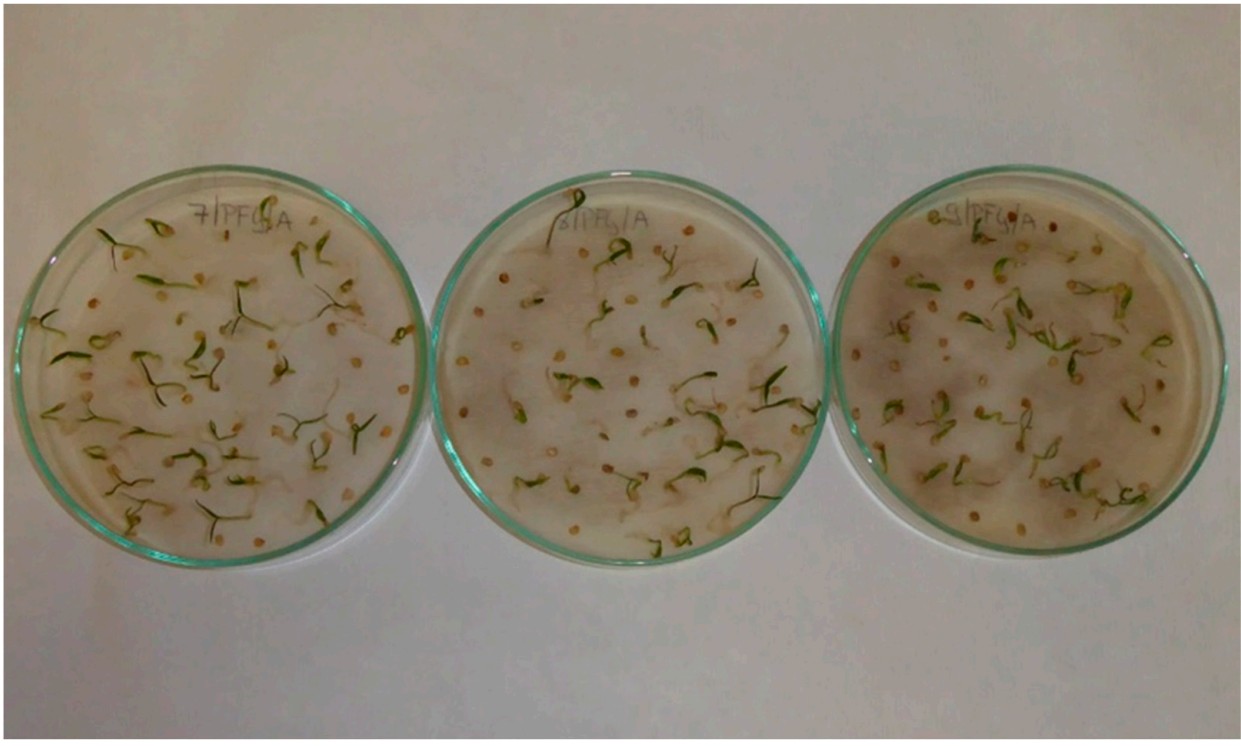

**Figure A2.** Pepper (*Capsicum anuum* L.) seedlings; treatments with aqueous extracts of black cumin (*Nigella sativa* L.) in 2.5% concentration (**left**), 5% concentration (**middle**) and 10% concentration (**right**).

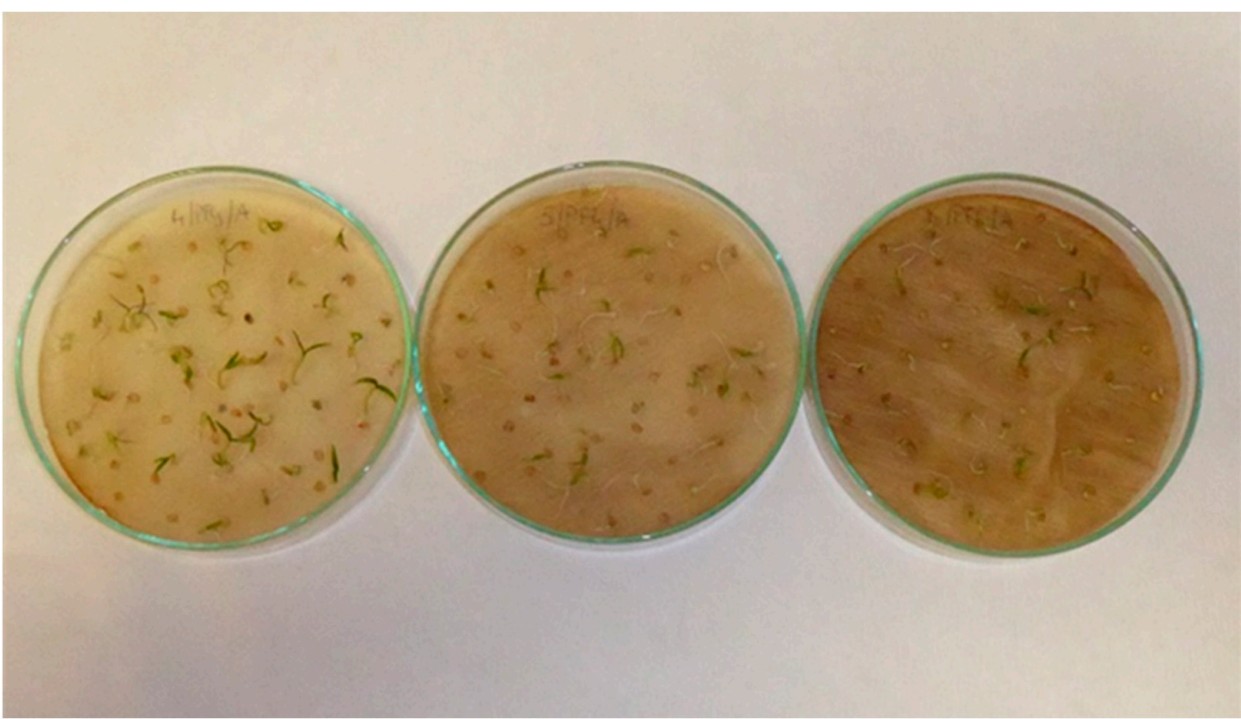

**Figure A3.** Pepper (*Capsicum anuum* L.) seedlings; treatments with aqueous extracts of wormwood herbs (*Artemisia absinthium* L.) in 2.5% concentration (**left**), 5% concentration (**middle**) and 10% concentration (**right**).

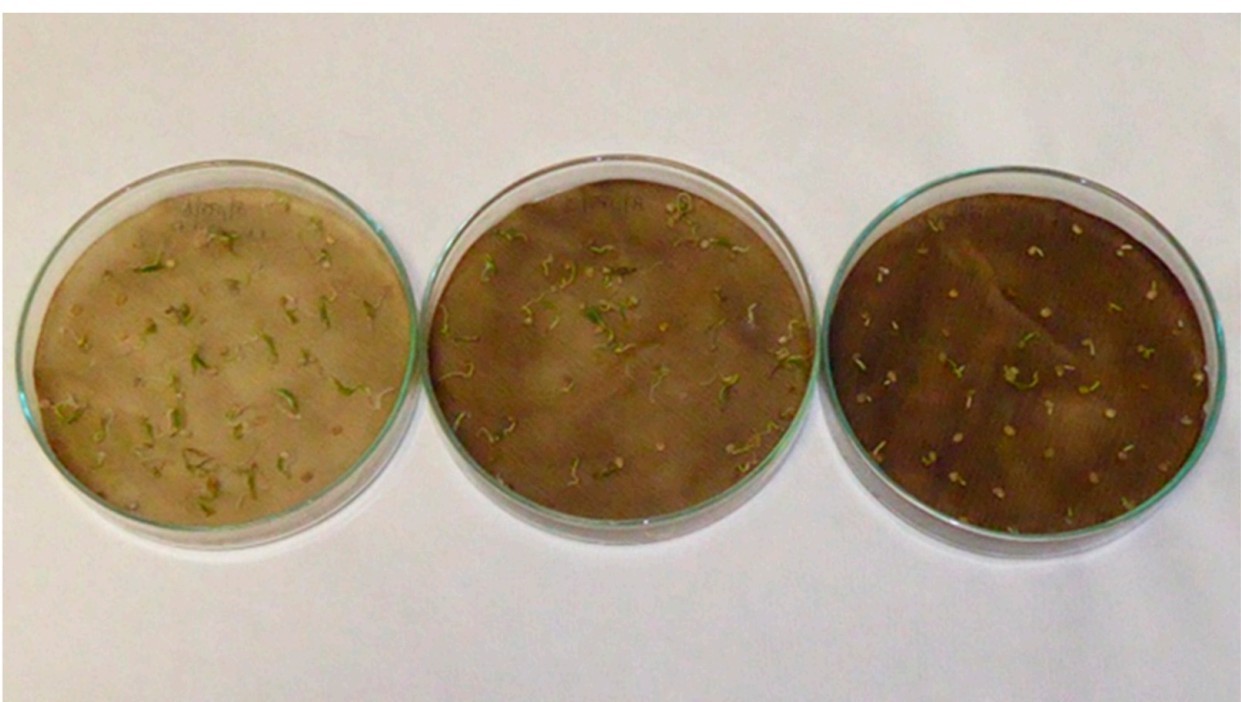

**Figure A4.** Pepper (*Capsicum anuum* L.) seedlings; treatments with aqueous extracts of nettle (*Urtica dioica* L.) in 2.5% concentration (**left**), 5% concentration (**middle**) and 10% concentration (**right**).

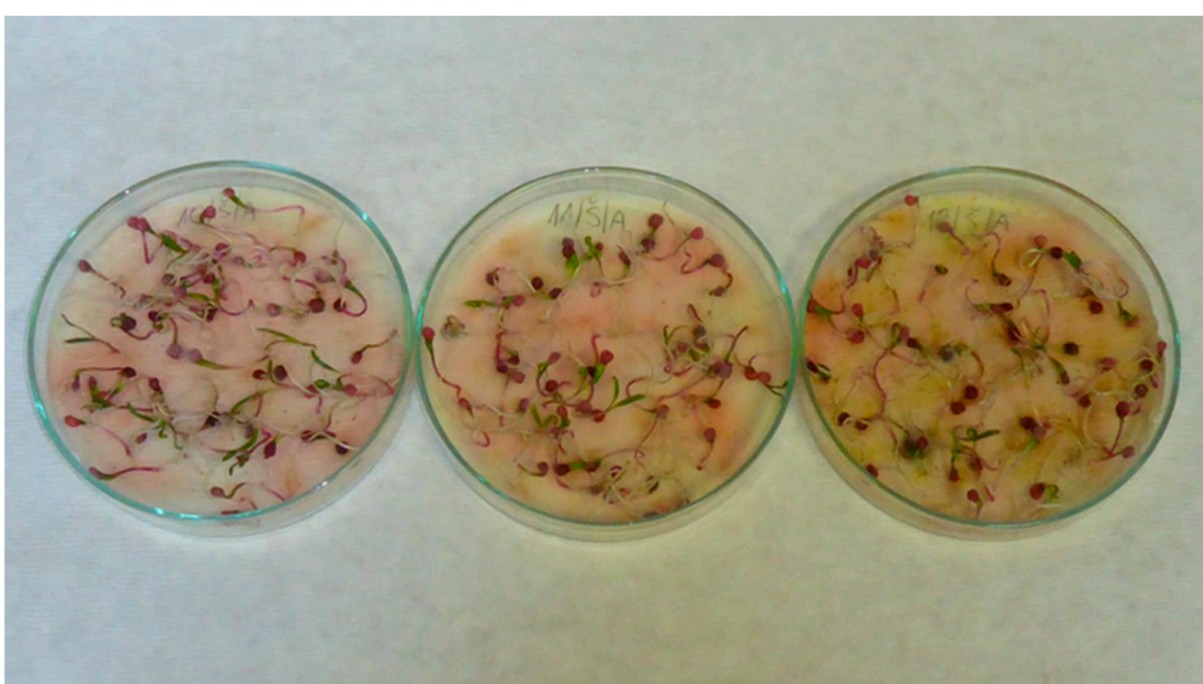

**Figure A5.** Spinach (*Spinacia oleracea* L.) seedlings; treatments with aqueous extracts of chia (*Salvia hispanica* L.) in 2.5% concentration (**left**), 5% concentration (**middle**) and 10% concentration (**right**).

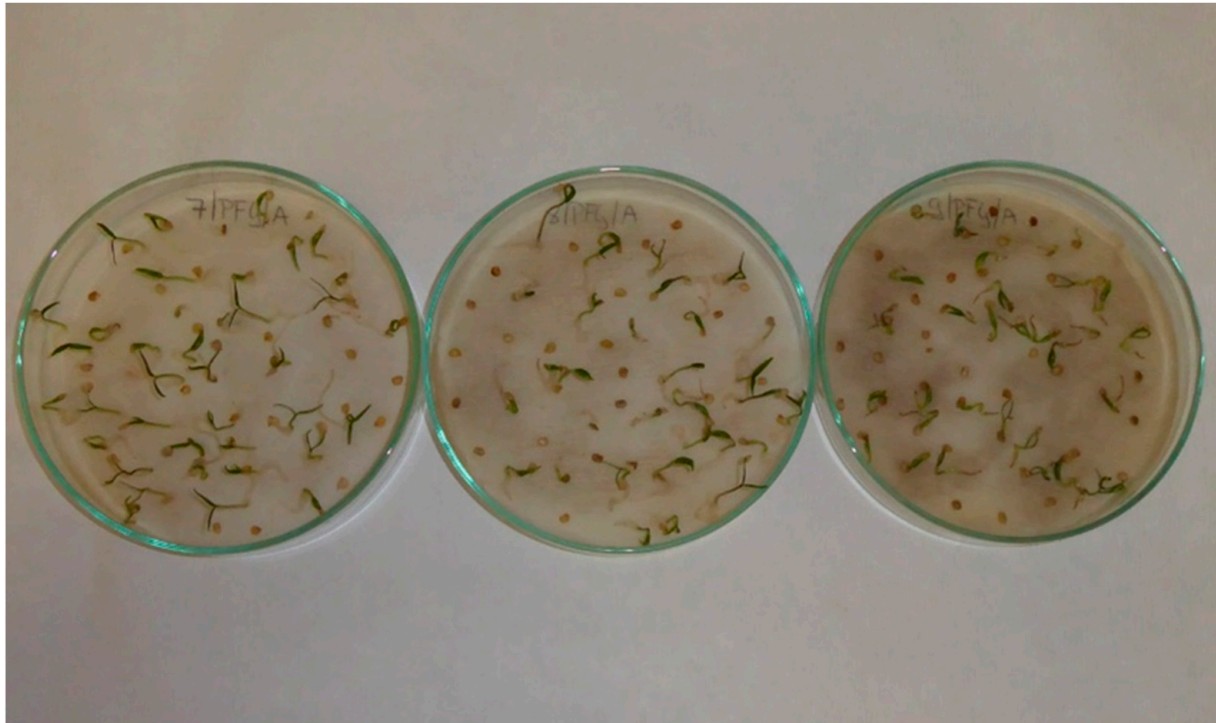

**Figure A6.** Spinach (*Spinacia oleracea* L.) seedlings; treatments with aqueous extracts of black cumin (*Nigella sativa* L.) in 2.5% concentration (**left**), 5% concentration (**middle**) and 10% concentration (**right**).

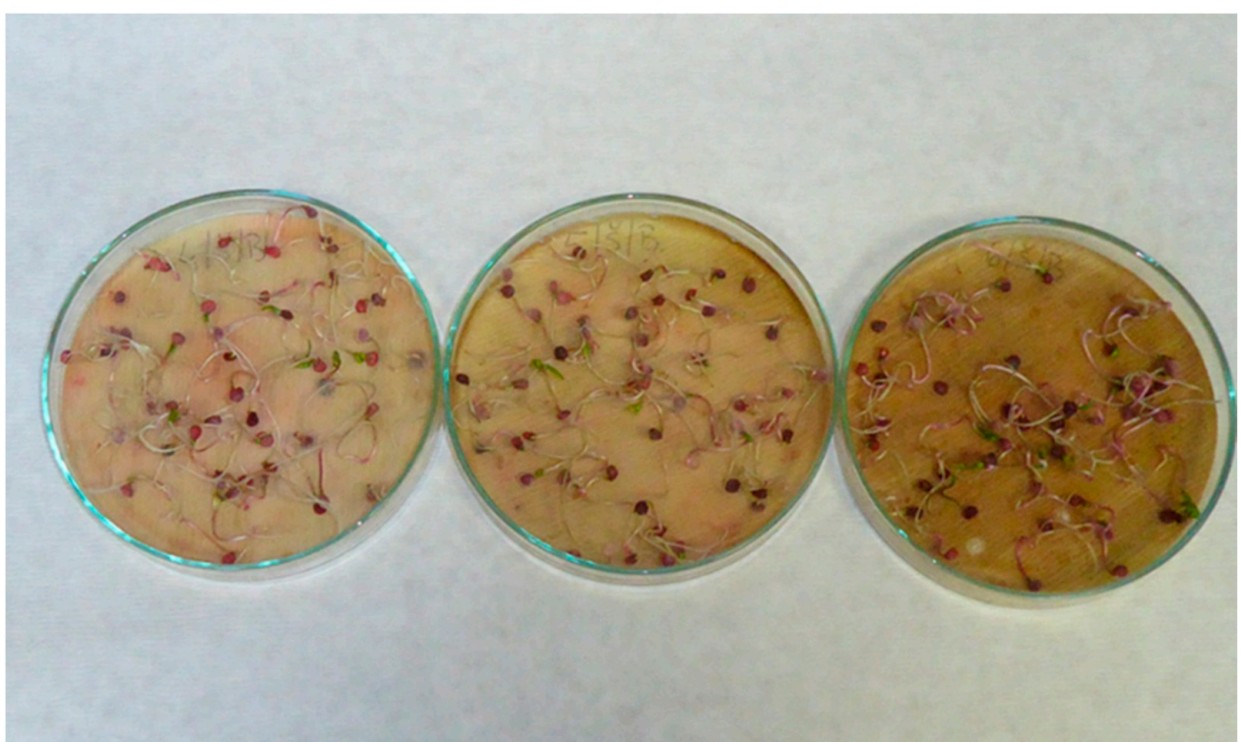

**Figure A7.** Spinach (*Spinacia oleracea* L.) seedlings; treatments with aqueous extracts of wormwood herbs (*Artemisia absinthium* L.) in 2.5% concentration (**left**), 5% concentration (**middle**) and 10% concentration (**right**).

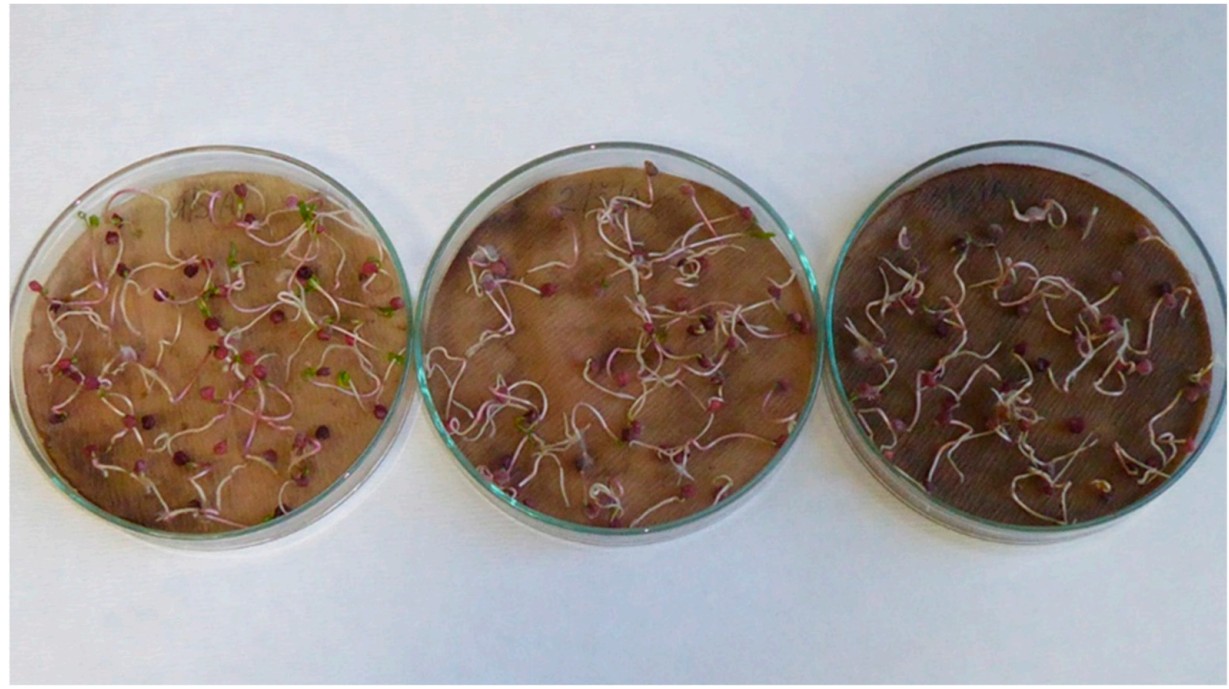

**Figure A8.** Spinach (*Spinacia oleracea* L.) seedlings; treatments with aqueous extracts of nettle (*Urtica dioica* L.) in 2.5% concentration (**left**), 5% concentration (**middle**) and 10% concentration (**right**).

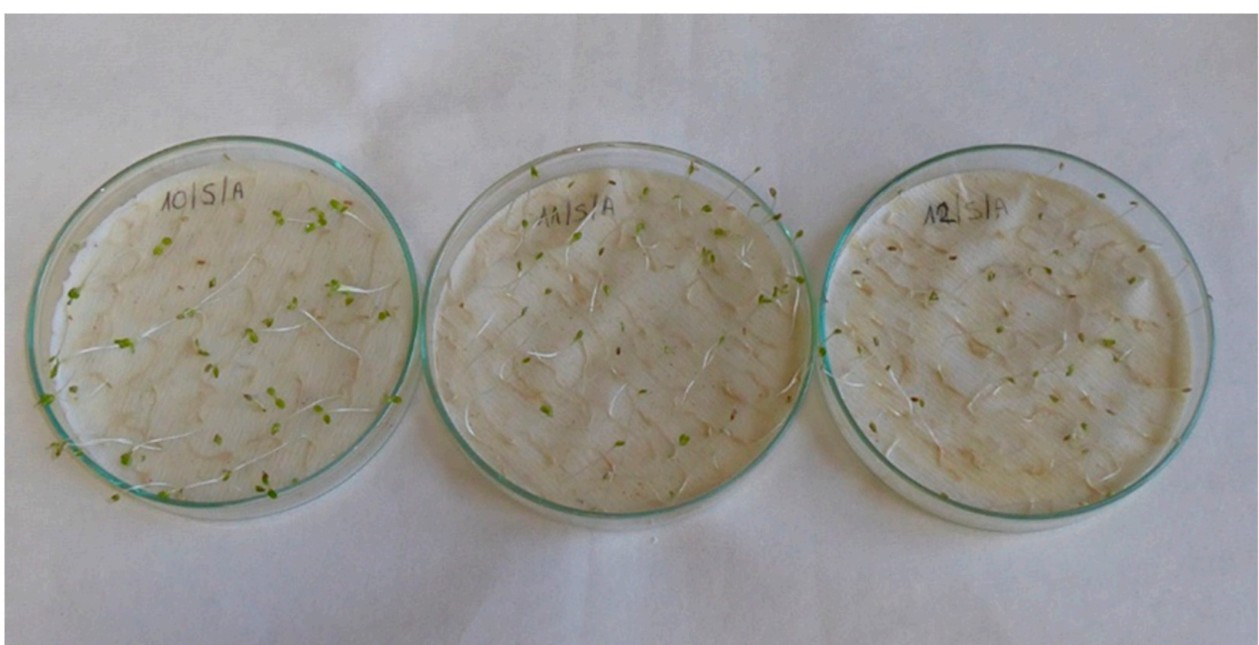

**Figure A9.** Lettuce (*Lactuca sativa* L.) seedlings; treatments with aqueous extracts of chia (*Salvia hispanica* L.) in 2.5% concentration (**left**) 5% concentration (**middle**) and 10% concentration (**right**).

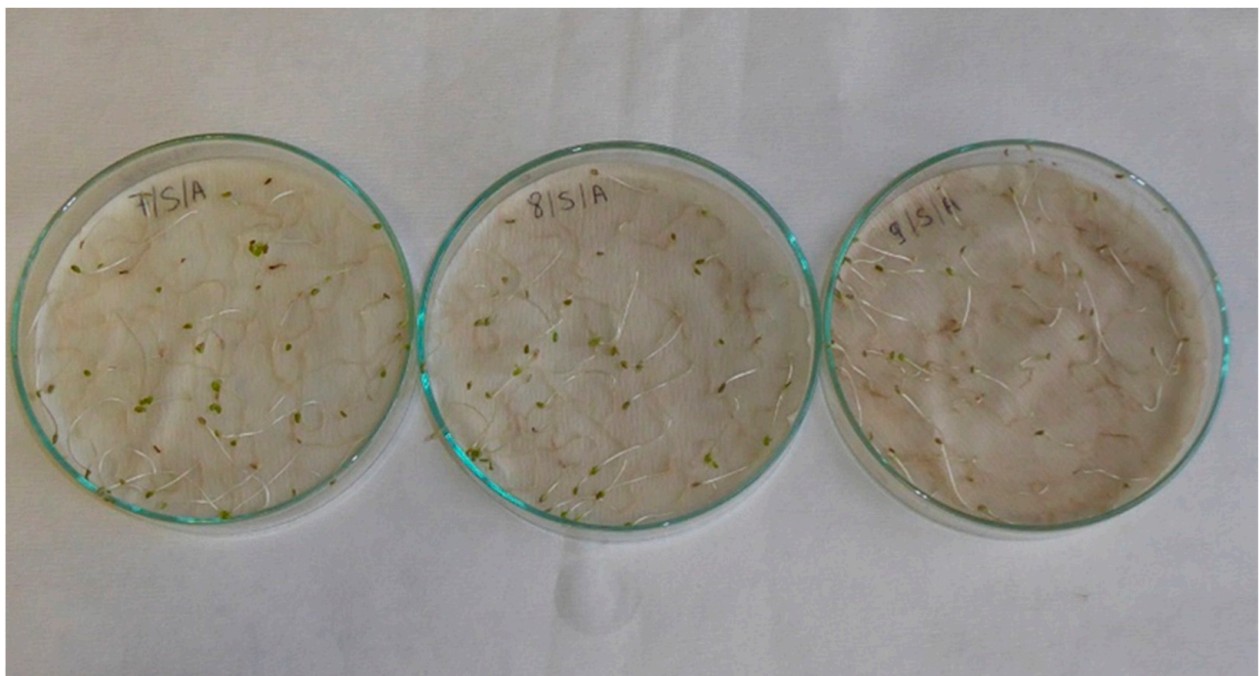

**Figure A10.** Lettuce (*Lactuca sativa* L.) seedlings; treatments with aqueous extracts of black cumin (*Nigella sativa* L.) in 2.5% concentration (**left**) 5% concentration (**middle**) and 10% concentration (**right**).

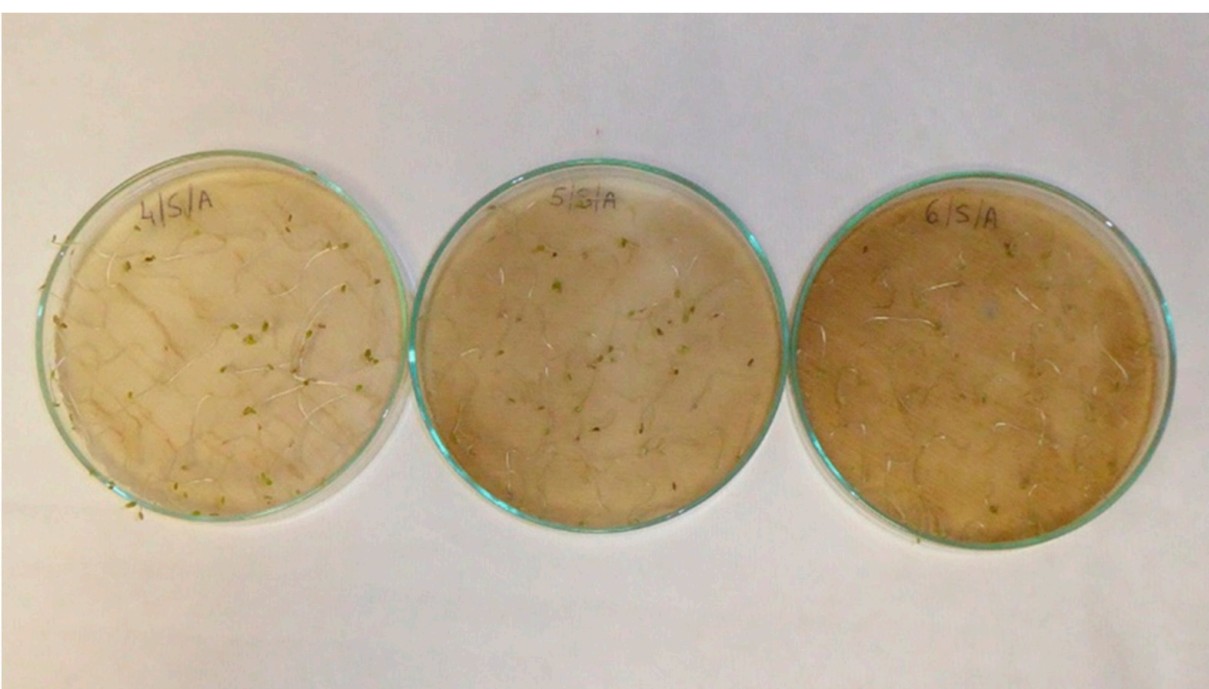

**Figure A11.** Lettuce (*Lactuca sativa* L.) seedlings; treatments with aqueous extracts of wormwood herbs (*Artemisia absinthium* L.) in 2.5% concentration (**left**) 5% concentration (**middle**) and 10% concentration (**right**).

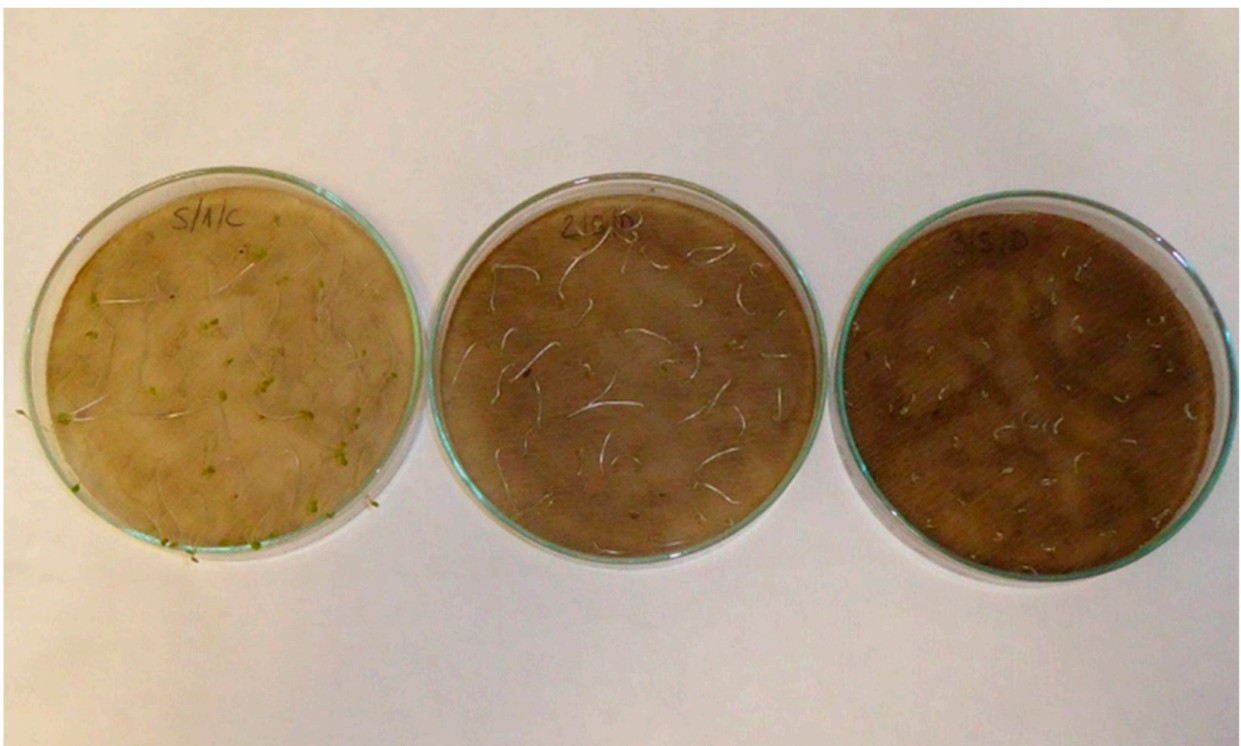

**Figure A12.** Lettuce (*Lactuca sativa* L.) seedlings; treatments with aqueous extracts of nettle (*Urtica dioica* L.) in 2.5% concentration (**left**), 5% concentration (**middle**) and 10% concentration (**right**).

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
