# Peer review of "Aqueous Extracts of Four Medicinal Plants and Their Allelopathic Effects on Germination and Seedlings: Their Morphometric Characteristics of Three Horticultural Plant Species"

_applsci, doi:10.3390/app13042258_

Round 1
Reviewer 1 Report
The authors undertook the Aqueous Extracts of Some Medicinal Plants and Their Allelopathic Effects on Germination and Seedlings Morphometric Characteristics of Some Horticultural Plant Species.
1. Which component is responsible for germination.
2. What about the chlorophyll and protein content.
3. What is the transport mechanism??? Here some references that help you
https://doi.org/10.1007/s10853-018-2107-9
https://doi.org/10.1039/C6EN00385K
4. Compare your results with existing similar materials.
5. It is seem difficult find novelty.
Author Response
Review #1
- Which component is responsible for germination.
A: Dear reviewer, thank you for this question. It is well-known there is many phytohormones which promote germination and also considering the allelopathic influence of determined secondary metabolites in plant aqueous extracts, and results of previous authors referred in the text, it is obvious that the secondary metabolites in higher content than 1 mgL-1, such as epicatechin, quinic acid, caffeic acid, esculetin, cinnamic acid, kaemferol, and gallic acids have the strongest allelopathic effect on seed germination. Thank you.
- What about the chlorophyll and protein content.
A: Dear reviewer, as the matter in fact, the chlorophyll and protein content were not the scope of this research. Namely, because of lack of references related to allelopathic influence of plant aqueous extracts on seed germination and morphometric characteristics of seedlings the aim of this study was to investigate the possible allelopathic influence of aqueous extracts obtained from plant species chia (Salvia hispanica L.), black cumin (Nigella sativa L.), wormwood (Artemisia absinthium L.), and nettle (Urtica dioica L.) on seed germination and morphometric characteristics of pepper (Capsicum annuum L.), spinach (Spinacia oleracea L.) and lettuce (Lactuca sativa L.) seedlings in laboratory conditions. Namely, these three species were chosen because of naturally huge variability of germination rate of them and because of huge variability of seedlings hypocotyl and radicle length. Additional explanation is now added in Introduction (lines 88-90).
- What is the transport mechanism??? Here some references that help you
https://doi.org/10.1007/s10853-018-2107-9
https://doi.org/10.1039/C6EN00385K
A: Dear reviewer, thank you for your advice. Yes, the references you suggested (number 35 and 36) just fit into Discussion chapter line 382. Thank you.
- Compare your results with existing similar materials.
A: Dear reviewer, the comparison with existing similar materials considering the allelopathic effects of various aqueous plant extracts have been done. Thank you.
- It is seem difficult find novelty.
A: Dear reviewer, yes I agree but that could be your impression in previous version of manuscript. However, after the changes we made, following your benevolent suggestions we assume that we clarified this issue.

Reviewer 2 Report
1) ALL UNITS given in the article should be written with a superscript. For example,...The mg/L must be mg L-1
2) rpm must be min-1
3) You should give the technical specifications of the Retsch GM 200
4) You must provie the technical specifications of the UPLC-ESI-MS/MS
5) from Table 5 to 13 … If there is a statistical difference, you should give the letter indicating the difference.
6) and if there is no statistical difference, you must see as “ns”
7) In the abstract section, it is necessary to introduce the work with 2 lines instead of entering directly as the aim.
8) Which program was used for statistical analysis?
Author Response
Review #2
1) ALL UNITS given in the article should be written with a superscript. For example,...The mg/L must be mg L-1
A: Dear reviewer, all units are now rewritten as suggested (lines 25, 30, 119, 175, 192, 204, 357, 360, and 362). Thank you.
2) rpm must be min-1
A: Dear reviewer, this is now corrected in line 103. Thank you.
3) You should give the technical specifications of the Retsch GM 200
A: Dear reviewer, this is now added in lines 102-103. Thank you.
4) You must provie the technical specifications of the UPLC-ESI-MS/MS
A: Dear reviewer, in section 2.2. “Phytochemical analyses” the type and producer of UPLC instrument are already stated (Agilent 1290, Agilent, Santa Clara, USA), as well as the type of linked triple quadropol: 6430 QqQ (Agilent). Moreover, UPLC operating conditions and column type are also described in this section, along with literature reference in which the method used for the qualitative and quantitative analysis of phenolic compounds in examined aqueous extracts is described in detail (Elez Garofulić et al. 2018; https://doi.org/10.1111/1750-3841.14368). Therefore, all necessary UPLC-ESI-MS/MS specifications have been provided in the manuscript. Thank you.
5) from Table 5 to 13 … If there is a statistical difference, you should give the letter indicating the difference.
A: Dear reviewer, yes I agree, we used the * and ** signs for significance at p<0.05 and 0.01, respectively. However, because of perceptiveness we changed that system of marking with letters, i.e., the values with different letters in superscription are significant. Thank you.
6) and if there is no statistical difference, you must see as “ns”
A: Dear reviewer, please see the changes in text and explanation below the tables. Thank you.
7) In the abstract section, it is necessary to introduce the work with 2 lines instead of entering directly as the aim.
A: Dear reviewer, we added an introducing sentence about allelopathy in the abstract (lines 13-14). Thank you.
8) Which program was used for statistical analysis?
A: Dear reviewer, for statistical analyses the Statistica TIBCO software was used, line 164. Thank you.

Reviewer 3 Report
Comments:
In the work “Aqueous Extracts of Some Medicinal Plants and Their Allelopathic Effects on Germination and Seedlings Morphometric Characteristics of Some Horticultural Plant Species” (applsci-2201382) by Erhatić et al, the authors have tried to investigate the possible allelopathic influence of aqueous extracts obtained from four plant species on the physiological performances of three plant species. Some interesting results showed that chia aqueous extract in concentration of 2.5% and 5% stimulate germination of spinach seeds and, 2.5% or even 10% stimulate the germination of lettuce seeds, and many other interesting results, which seems lettuce is the easiest species get the benefits from aqueous extracts. This may need to be highlighted in the abstract and discussion. The manuscript is generally well-written and may be useful for agriculture. My concerns may need to address in their revised manuscript:
1. The theory of allelopathy should be consolidated in the background. The current version has a very limited background.
2. Why there occurs a species-specific response to aqueous extracts? And most importantly, are there any significant differences among the species studied? A two-way ANOVA or GLMM procedure is needed to reveal.
3. The results of germination tests, and morphometric analyses of seedlings should be illustrated by graphs, rather than tables.
4. The narrative of the abstract can be consistent with the full text, i.e., the order of appearance of the plant species.
5. Chang the title into “Aqueous Extracts of Four Medicinal Plants and Their Allelopathic Effects on Germination and Seedlings Morphometric Characteristics of Three Horticultural Plant Species”, which is more specific.
Small points:
Line 81: delete “growth”.
Line 130: “for 14 days, 21 days for spinach and 7 days for lettuce”, why did you choose different days?
Line 361: Do you mean “biomass”?
Line 384-386: delete, or move to the introduction.
Author Response
Review #3
- The theory of allelopathy should be consolidated in the background. The current version has a very limited background.
A: Dear reviewer, I agree with you 100 %. In the first version of manuscript, I did that. Nevertheless, when the manuscript was finished, we noticed a huge number of pages almost like the review paper. So we decided to cut the text, but we left all important references where the theory of allelopathy is explained, following the historical order. References # 1, 10, 16, 41. Thank you.
- Why there occurs a species-specific response to aqueous extracts? And most importantly, are there any significant differences among the species studied? A two-way ANOVA or GLMM procedure is needed to reveal.
A: Dear reviewer, methodologically and statistically speaking you’re right. However, as you surely know these plant species have different rate of seed germination by itself. Namely, the minimal seed germination rate, prescribed by the seed testing authorities of each country vary from 65 % for pepper seeds till 75 % for lettuce seeds. Thus, their germination rates are not comparable. Respecting these reasons, the differences are quite obvious without two-way ANOVA and comparison of means between species. Moreover, in the contrary, the experts for seed production will surely wonder “why these authors compared the differences between these species, which are completely incomparable”. Thank you.
- The results of germination tests, and morphometric analyses of seedlings should be illustrated by graphs, rather than tables.
A: Dear reviewer, we agree with you, but your kind suggestion is somehow in collision with reviewer #2. So we replaced the * and ** signs for significance level of p<0.05 and 0.01, respectively, with different letters and not significant differences are signed with abbreviation of n.s., as previous reviewer requested. Nevertheless, as you surely know, the ununiformed germination of pepper, spinach and lettuce seeds are the real issue in production of seedlings of these species. So, the experts for seed production as well as for production of seedlings would really like to see the table with numbers instead of graphical chart. So, we kindly ask you to accept our reasons to leave the results in tables instead of exposing them in charts. Thank you.
- The narrative of the abstract can be consistent with the full text, i.e., the order of appearance of the plant species.
A: Dear reviewer, the order of appearance of the plant species is now corrected through all manuscript (lines 15-17, 85-86, 154-155, 323-324, 402-403). Thank you.
- Chang the title into “Aqueous Extracts of Four Medicinal Plants and Their Allelopathic Effects on Germination and Seedlings Morphometric Characteristics of Three Horticultural Plant Species”, which is more specific.
A: Dear reviewer, suggested title is now accepted. Thank you.
Small points:
Line 81: delete “growth”.
A: Dear reviewer, this is now deleted (line 87). Thank you.
Line 130: “for 14 days, 21 days for spinach and 7 days for lettuce”, why did you choose different days?
A: Dear reviewer, the number of days for seeds germination tests i.e., the number of days for each counting of germinated seeds, for every cultivated plant species is proposed by ISTA methodology we referred in further text in the same chapter reference 21. Nevertheless, because of your remark we put the reference 21 on the end of line 139. Thank you.
Line 361: Do you mean “biomass”?
A: Dear reviewer, “mass” is now corrected to “biomass” (line 372). Thank you.
Line 384-386: delete or move to the introduction.
A: Dear reviewer, this is now moved to the Introduction (lines 66-69). Thank you.

Reviewer 4 Report
The manuscript entitled “Aqueous Extracts of Some Medicinal Plants and Their Allelopathic Effects on Germination and Seedlings Morphometric Characteristics of Some Horticultural Plant Species” authored by Renata Erhatić, Dijana Horvat, Zoran Zorić, Maja Repajić, Tanja Jović, Martina Herceg, Matea Habuš and Siniša Srečec, presents the investigation of various plant extracts in terms of their allelopathic effects on germination and seedlings morphometric characteristics of pepper (Capsicum annuum L.), spinach (Spinacia oleracea L.) and lettuce (Lactuca sativa L.).
The manuscript is nicely written, well organized and contains quite interesting information of general importance for agriculture. In my opinion, it should be accepted for publication after some minor corrections listed below.
- line 21: the abbreviation UPLC-ESI-MS/MS should be defined here.
- lines 20-21: “…and the qualitative and quantitative analysis of phenolic compounds in aqueous extracts was determined using UPLC-ESI-MS/MS.” Instead of “ was determined” it should be “were carried out” or “were done”…the analysis is done, not determined.
- line 44: “and” should be corrected to “or”.
Author Response
Review #4
The manuscript is nicely written, well organized and contains quite interesting information of general importance for agriculture. In my opinion, it should be accepted for publication after some minor corrections listed below.
- line 21: the abbreviation UPLC-ESI-MS/MS should be defined here.
A: Dear reviewer, the abbreviation UPLC-ESI-MS/MS is now defined in line 23. Thank you.
- lines 20-21: “…and the qualitative and quantitative analysis of phenolic compounds in aqueous extracts was determined using UPLC-ESI-MS/MS.” Instead of “ was determined” it should be “were carried out” or “were done”…the analysis is done, not determined.
A: Dear reviewer, this is now corrected to “were done” (line 22). Thank you.
- line 44: “and” should be corrected to “or”.
A: Dear reviewer, we changed “and” to “or” (line 46). Thank you.

Round 2
Reviewer 1 Report
Accept
Reviewer 3 Report
The authors seem to have done good work in revising the manuscript I think it can be accepted for publication now. Cheers.